# Failure to receive prescribed imaging is associated with increased early mortality after injury in Cameroon

Matthew Driban[1]*, Fanny N. Dissak-Delon[2], Melissa Carvalho[3], Mbiarikai Mbianyor[3], Georges A. Etoundi-Mballa[4], Thompson Kingue[5], Richard L. Njock[6], Daniel N. Nkusu[7], Jean-Gustave Tsiagadigui[8], Juan C. Puyana[9], Catherine Juillard[3], Alain Chichom-Mefire[10], S. Ariane Christie[3]

1 University of Pittsburgh School of Medicine, Pittsburgh, Pennsylvania, United States of America, 2 University of Bamenda Faculty of Health Sciences, Bamenda, Cameroon, 3 University of California–Los Angeles Department of Surgery, Program for the Advancement of Surgical Equity, Los Angeles, California, United States of America, 4 Ministry of Public Health of Cameroon Department of Disease Epidemic and Pandemic Control, Yaounde, Centre Region, Cameroon, 5 The Limbe Regional Hospital Hospital Administration, Limbe, Cameroon, 6 The Laquintinie Hospital of Douala Hospital Administration, Douala, Cameroon, 7 The Catholic Hospital of Pouma Hospital Administration, Pouma, Cameroon, 8 The Edea Regional Hospital Hospital Administration, Edea, Cameroon, 9 University of Pittsburgh Department of Trauma and Critical Care, Pittsburgh, Pennsylvania, United States of America, 10 University of Buea Faculty of Health Sciences, Buea, Cameroon

* mdriban75@gmail.com

**Data Availability Statement:** De-identified data provided in S1 Data.

## Abstract

Despite having the highest rates of injury-related mortality in the world, trauma system capacity in sub-Saharan Africa remains underdeveloped. One barrier to prompt diagnosis of injury is limited access to diagnostic imaging. As part of a larger quality improvement initiative and to assist priority setting for policy makers, we evaluated trauma outcomes among patients who did and did not receive indicated imaging in the Emergency Department (ED). We hypothesize that receiving imaging is associated with increased early injury survival. We evaluated patterns of imaging performance in a prospective multi-site trauma registry cohort in Cameroon. All trauma patients enrolled in the Cameroon Trauma Registry (CTR) between 2017 and 2019 were included, regardless of injury severity. Patients prescribed diagnostic imaging were grouped into cohorts who did and did not receive their prescribed study. Patient demographics, clinical course, and outcomes were compared using chi-squared and Kruskal-Wallis tests. Multivariate logistic regression was used to explore associations between radiologic testing and survival after injury. Of 9,635 injured patients, 47.5% (4,574) were prescribed at least one imaging study. Of these, 77.8% (3,556) completed the study (COMPLETED) and 22.2% (1,018) did not receive the prescribed study (NC). Compared to COMPLETED patients, NC patients were younger (p = 0.02), male (p<0.01), and had markers of lower socioeconomic status (SES) (p<0.01). Multivariate regression adjusted for age, sex, SES, and injury severity demonstrated that receiving a prescribed study was strongly associated with ED survival (OR 5.00, 95% CI 3.32–7.55). Completing prescribed imaging was associated with increased early survival in injured Cameroonian patients. In a

**Funding:** The Cameroon Trauma Registry was supported by University of California San Francisco (CJ) and Los Angeles Departments of Surgery (CJ) research funding, NIH R21TW010453 grant (CJ, ACM), the University of Pittsburgh School of Medicine Class of 1972 (MD). The funders had no role in study design, data collection and analysis, decision to publish, or preparation of the manuscript.

**Competing interests:** The authors have declared that no competing interests exist.

resource-limited setting, subsidizing access to diagnostic imaging may be a feasible target for improving trauma outcomes.

## Introduction

Trauma care in developed health systems relies on diagnostic imaging to help facilitate rapid diagnosis and aid in priority-setting for patients with multiple injuries [1–4]. Radiographs, extended focused assessment with sonography for trauma (eFAST), and multidetector computed tomography (CT) are standard components of the trauma assessment taught in Advanced Trauma Life Support (ATLS) and have resulted in improved trauma outcomes [5–9]. Despite bearing a disproportionate amount of the global injury burden, low- and middle-income countries (LMICs) have relatively few high-quality studies addressing diagnostic capacity for trauma. Limited available data, including a recent scoping review of imaging capacity for cancer, describe critical shortages in equipment and personnel in LMICs [10]. For example, there is only one CT-scanner available per one-million LMIC inhabitants, compared to 40 per one-million high-income country inhabitants, and a similar disparity in the availability of radiologists. Restricted access to imaging may delay diagnosis, which is known to confer increased rates of preventable trauma death [11]. However, due to variability in LMIC trauma systems, characterizing associations between imaging access and patient outcomes remains a critical knowledge gap in most LMICs. Understanding context-specific diagnostic imaging care patterns is necessary to identify high-impact targets for trauma systems improvements in environments where resources are scarce. However, many LMICs lack sufficiently granular data collection infrastructure to provide actionable data.

Cameroon is a mixed Francophone-Anglophone Central African nation with an estimated annual injury mortality rate that exceeds trauma mortality in the United States by 40% [12,13]. Its healthcare system includes no formal prehospital care, scant medical record keeping, and a fee-for-service model, with less than 7% of the entire population covered by health insurance [14]. A 2017 report generated by the not-for-profit organization Rad-Aid estimated that there are between 100 and 150 medical imaging centers serving a population of 21 million inhabitants [15]. While reportedly at least one government-funded imaging center in each of Cameroon's 10 geopolitical regions offers computed tomography, imaging capacity is largely concentrated in three major urban areas: Yaoundé, Douala, and Bafoussam. Over 80% of Cameroonian imaging centers are limited to radiographs and/or ultrasound. Most existing equipment is donated, refurbished, or has been in use for over 10 years [15].

Since 2015, the Cameroon Trauma Registry (CTR) has collected prospective data on injured Cameroonian patients with a goal of identifying high impact targets to improve trauma care [16,17]. Failure to diagnose critical injury, like hemorrhagic shock, in a timely manner has been implicated as a common root cause of preventable deaths by both the Cameroonian National Trauma Quality Improvement Committee and trauma death analysis from the national Cameroon Trauma Registry (CTR) [18]. To develop a context-tailored, data-driven approach to reducing diagnostic delays in Cameroon, in this study we characterized diagnostic imaging utilization patterns in a prospective, multicenter trauma registry cohort. Our aim was to characterize receipt of diagnostic imaging and compare outcomes between patients who did and did not receive indicated imaging in an LMIC trauma population. We hypothesized that receiving prescribed diagnostic imaging early in care is associated with increased survival after injury in Cameroon.

## Materials and methods

### Data source

We analyzed radiology utilization patterns in a Cameroonian trauma cohort using prospective registry data from the previously described CTR. The CTR was created in accordance with WHO Guidelines for Injury Surveillance and adapted to the local context and interest [17,19,20]. Trauma registrars followed all injured patients from presentation to hospital discharge. Data was collected on securely kept paper registry forms which were then entered into an encrypted cloud-based server [21,22].

### Ethics statement

The ongoing registry is managed jointly by the University of California, Los Angeles (UCLA) and the Cameroonian Ministry of Public Health. Ethical approval for the CTR is maintained through the Cameroon National Ethics Committee and UCLA institutional review boards. Informed oral consent to participate in the study was obtained by trained research assistants from patients or surrogates using an institutional review board (IRB)-approved oral consent script. Consent for minors was obtained through their guardian and oral assent was obtained from minors older than 7 years of age to the permission provided by their guardians for participation. Patients/surrogates were given the opportunity to ask questions and were assured that their participation or not in the study is voluntary and has no effect on their medical care. If obtained, consent was documented, and information was only collected on patients who gave their consent to participate. Informed verbal consent was obtained from the study participants due to variable rates of patient literacy because the study involves minimal risk with no procedures for which written consent is normally required outside the research setting.

### Study setting

The CTR collected data on all injured patients presenting to four trauma hospital between 2017 and 2019: Laquintinie Hospital of Doula (HLD), Limbe Regional Hospital (HRL), Catholic Hospital of Pouma (HCP), and Regional Hospital Annex of Edea (HRE).

### Data collection and analysis

CTR data was extracted on all patients presenting with injuries during the study period, regardless of age, disposition, or injury severity. To characterize diagnostic capacity, we identified all injured patients who were prescribed imaging as part of their clinical care. For trauma patients prescribed imaging, we compared clinical care patterns and outcomes between cohorts that received imaging (COMPLETED) and those who did not receive their prescribed study (NC). Data analyzed included: patient demographics, injury characteristics, vital signs, diagnostic studies ordered and performed, treatments ordered and performed, reasons that prescribed studies and treatments were not performed, clinical outcomes, and patient disposition through hospital discharge. As in prior studies, access to liquid petroleum gas (LPG) was used as a surrogate indicator of higher socioeconomic status (SES) [17,23,24]. Injury severity was reported using Highest Estimated Abbreviated Injury Scale (HEAIS). Clinical assessment of injury severity using HEAIS strongly predicts hospital mortality and far exceeds conventional scoring systems (Revised Trauma Score, Kampala Trauma Score) in feasibility [25].

Descriptive statistics were reported as proportions and frequencies for categorical variables, mean and standard deviation for normally distributed numeric variables, and median and interquartile range (IQR) for non-parametric continuous and ordinal variables. Cohort demographics and clinical outcomes, including survival, access to medical services, admission, and

inability to pay, were compared using chi-squared tests for categorical variables and Kruskal-Wallis for numeric variables. Multivariate logistic regression was used to explore associations between radiologic testing and clinical outcomes after injury. For all statistic tests, an alpha level of 0.05 was considered significant. Data was analyzed using Stata16 [26].

## Results

### Study cohort

Between 2017–19, CTR data was collected on 9,635 injured patients. The overall cohort was male-predominant (70.3%) with a median age of 30-years-old, reflecting a standard trauma population. The most common injury mechanism was road traffic injury (RTI) (55.8%).

### Imaging completion cohorts

Of 9,635 injured patients, 47.5% (n = 4,574) were prescribed at least one imaging study in the ED (Fig 1). Among patients prescribed imaging, 22.2% never received the prescribed study (NC, n = 1,018), while 77.8% received their prescribed imaging (COMPLETED, n = 3,556). NC patients were younger (p = 0.02), more commonly male (p<0.01), and had markers of lower SES (p<0.01, Table 1). COMPLETED patients also had higher rates of past medical contact (p<0.01).

### Diagnostic capacity

Successful completion of prescribed diagnostic imaging varied by study site, ranging from 50–100% (Fig 2). The most frequently reported reasons for not receiving a prescribed study included inability to pay (n = 498, 46.2%), lack of functional equipment (n = 220, 20.4%), patient preference (n = 175, 16.2%), patient departure against medical advice (n = 136, 12.6%), and lack of available staff (n = 57, 5.3%). A total of 3,682 studies were performed. Radiography was the most commonly performed imaging modality (n = 3,215, 87.3%) followed by CT (n = 388, 10.5%), ultrasound (n = 76, 2.1%), and magnetic resonance imaging (MRI) (n = 3,

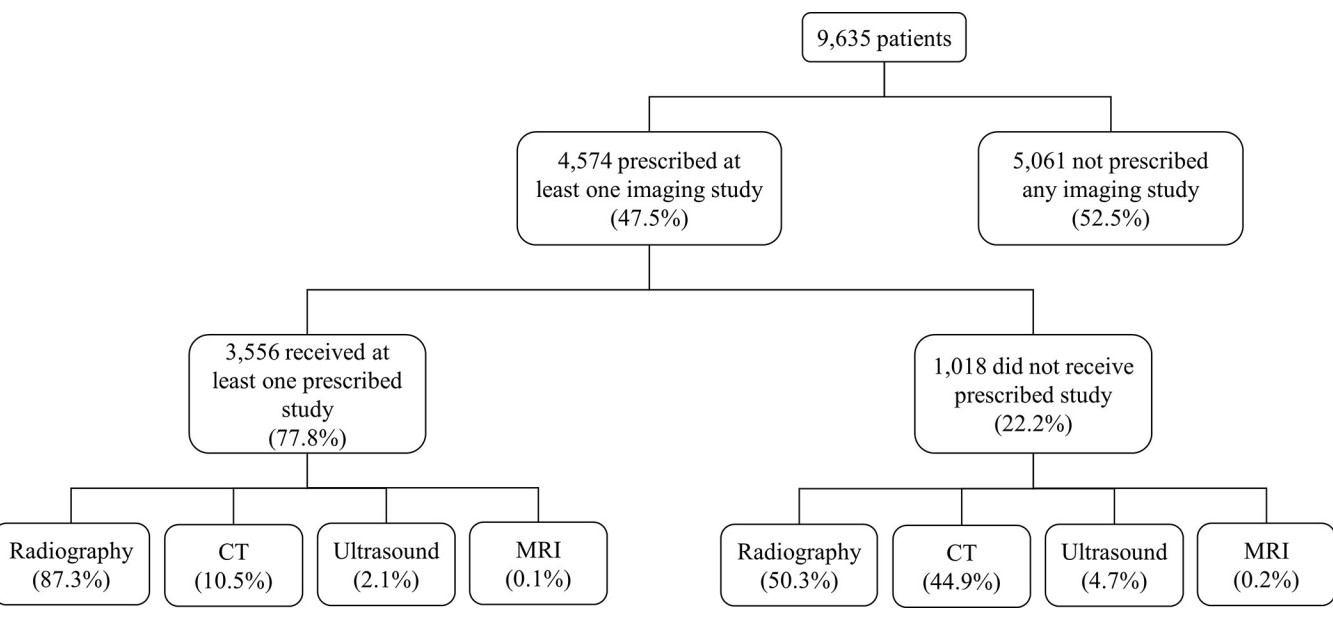

**Fig 1. Imaging completion cohorts.**

**Table 1. Demographics of injured patients.**

|  | COMPLETED Patients (n = 3,556) | NC Patients (n = 1,018) | p-value |
|---|---|---|---|
| Age, median (IQR), years | 33 (24–43) | 31 (23–40) | 0.02 |
| Male, n (%)* | 2,571 (72.7%) | 784 (77.2%) | <0.01 |
| Access to Liquid Petroleum Gas | 2,219 (62.4%) | 534 (52.5%) | <0.01 |
| Owns Cellphone, n (%) | 3,374 (95.2%) | 915 (90.0%) | <0.01 |
| Past Medical History, n (%) | 368 (10.4%) | 63 (6.2%) | <0.01 |

NC, not completed; IQR, interquartile range.

0.1%). Pathology was demonstrated in 59.0% of completed studies (n = 2,171). The distribution of imaging modalities differed significantly between cohorts (p<0.01), with a significantly higher proportion of incomplete scans being classified as CT or ultrasound.

### Patient presentation and injury characteristics

Patient presentation and injury characteristics differed between cohorts (Table 2). Presenting vitals did not differ between cohorts, but COMPLETED patients had more severe injuries (p<0.01) and higher rates of polytrauma (28.2% vs 24.8%, p = 0.03) than NC patients. NC patients had higher rates of primary survey abnormalities (51.3% vs 47.0%, p = 0.02) and active bleeding (65.9% vs 60.0%, p<0.01). COMPLETED patients had higher rates of extremity (p<0.01), chest (p<0.01), and spine (p = 0.05) injuries while NC patients had higher rates of face (p<0.01), head/neck (p<0.01), and abdominal (p = 0.02) injuries. Fall and cut/stab

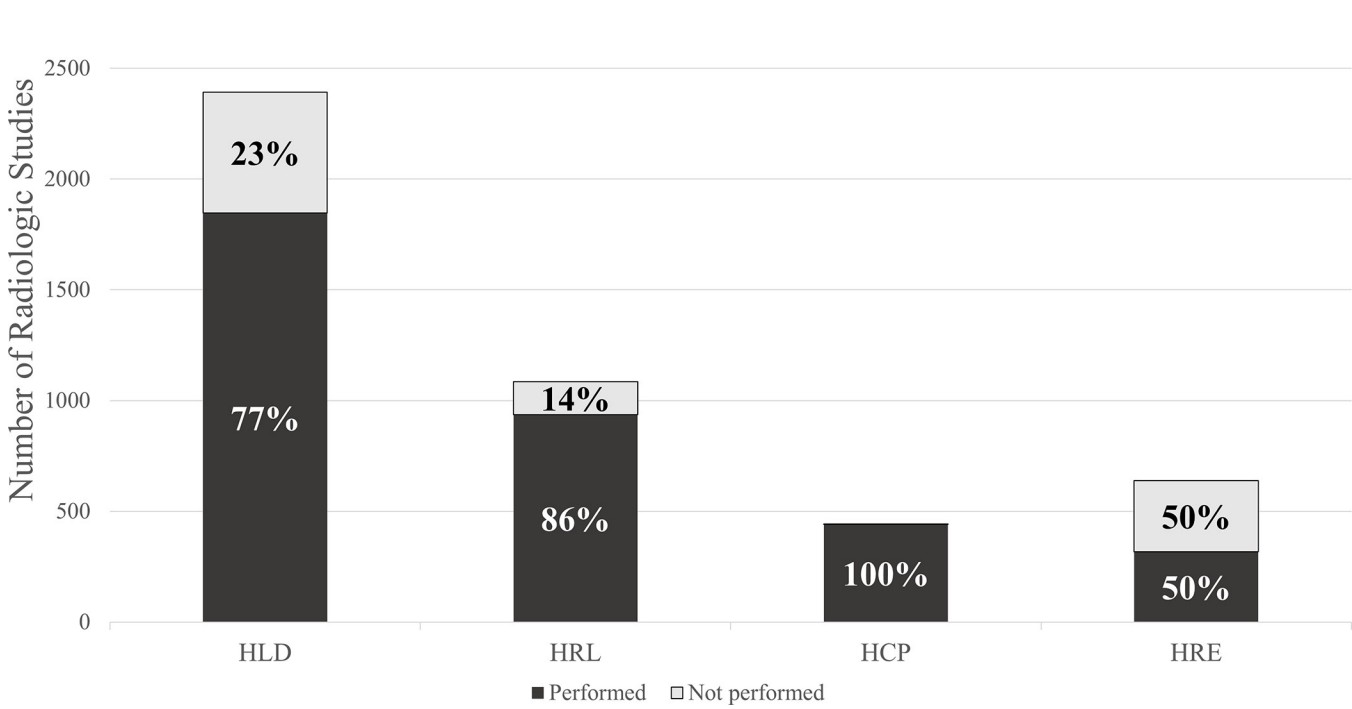

**Fig 2. Imaging completion by hospital.** HLD is a central hospital in Cameroon's capital city, Yaoundé. HRL, HCP, and HRE are regional/district hospitals in the Littoral and Southwest Regions of Cameroon.

**Table 2. Physiologic parameters and injury characteristics of injured patients.**

| | | COMPLETED Patients (n = 3,556) | NC Patients (n = 1,018) | p-value |
|---|---|---|---|---|
| Presenting Vital Signs, mean (SD) | | | | |
| | Heart Rate | 87.0 (16.6) | 87.2 (19.1) | 0.81 |
| | MAP | 94.8 (15.7) | 94.2 (16.5) | 0.27 |
| | Respiratory Rate | 22.0 (5.5) | 21.6 (4.7) | 0.15 |
| HEAIS, median (IQR) | | 3 (2–3) | 2 (2–3) | <0.01 |
| Polytrauma, n (%) | | 1,002 (28.2%) | 252 (24.8%) | 0.03 |
| Primary Survey Abnormality, n (%) | | 1,673 (47.0%) | 522 (51.3%) | 0.02 |
| Active Bleeding, n (%) | | 2,135 (60.0%) | 671 (65.9%) | <0.01 |
| Injury Location, n (%) | | | | |
| | Extremity | 2,730 (76.8%) | 598 (58.7%) | <0.01 |
| | Face | 1,016 (28.6%) | 395 (38.8%) | <0.01 |
| | Head/Neck | 816 (22.9%) | 413 (40.6%) | <0.01 |
| | Chest | 639 (18.0%) | 102 (10.0%) | <0.01 |
| | Pelvis | 177 (5.0%) | 37 (3.6%) | 0.07 |
| | Spine | 129 (3.6%) | 24 (2.4%) | 0.05 |
| | Abdomen | 127 (3.6%) | 53 (5.2%) | 0.02 |
| Injury Mechanism, n (%) | | | | |
| | Road Traffic Injury | 2,307 (64.9%) | 765 (75.1%) | <0.01 |
| | Fall | 602 (16.9%) | 95 (9.3%) | <0.01 |
| | Non-Fall Blunt Force | 240 (6.7%) | 103 (10.1%) | <0.01 |
| | Cut/Stab | 230 (6.5%) | 31 (3.0%) | <0.01 |

NC, not completed; SD, standard deviation; HEAIS, highest estimated abbreviated injury severity; IQR, interquartile range.

mechanisms were more common among COMPLETED patients (p<0.01). RTIs and non-fall blunt force mechanisms were more common among NC patients (p<0.01).

## Care utilization patterns

Multivariate logistic regression adjusted for age, sex, SES, and injury severity showed that completing prescribed imaging was strongly associated with increased rates of hospital receipt of additional medical services (OR 1.55, 95% CI 1.40–1.72), including procedures, consultations, and lab testing (Fig 3). COMPLETED patients had significantly higher rates of receipt of analgesics, antibiotics, split casts, reduction, and consults to general/orthopedic surgery (p<0.01). NC patients received a higher rate of non-general/orthopedic surgery consults (p<0.01), including neurosurgery, plastic surgery, vascular surgery, and others. Overall, both cohorts received medications at a similar rate (94.8% vs 90.9%, p = 0.51). NC patients reported a significantly higher rate of cost interference with care (60.5%) compared to COMPLETED patients (29.7%) (p<0.01), despite facing a lower average cost of care (p<0.01).

## Emergency department dispositions

Emergency department disposition varied by imaging completion status. COMPLETED patients had higher rates of admission (25.0% vs 12.7%, p<0.01) and surgery (4.2% vs. 0.8%, p<0.01). NC patients left against medical advice (35.0% vs. 21.3%, p<0.01) or died (5.6% vs. 1.3%, p<0.01) more frequently. When adjusted for age, sex, SES, and injury severity, multivariate logistic regression demonstrated that completing a prescribed radiology study was independently associated with ED survival (OR 5.00, 95% CI 3.32–7.55) (Table 3). Median time to

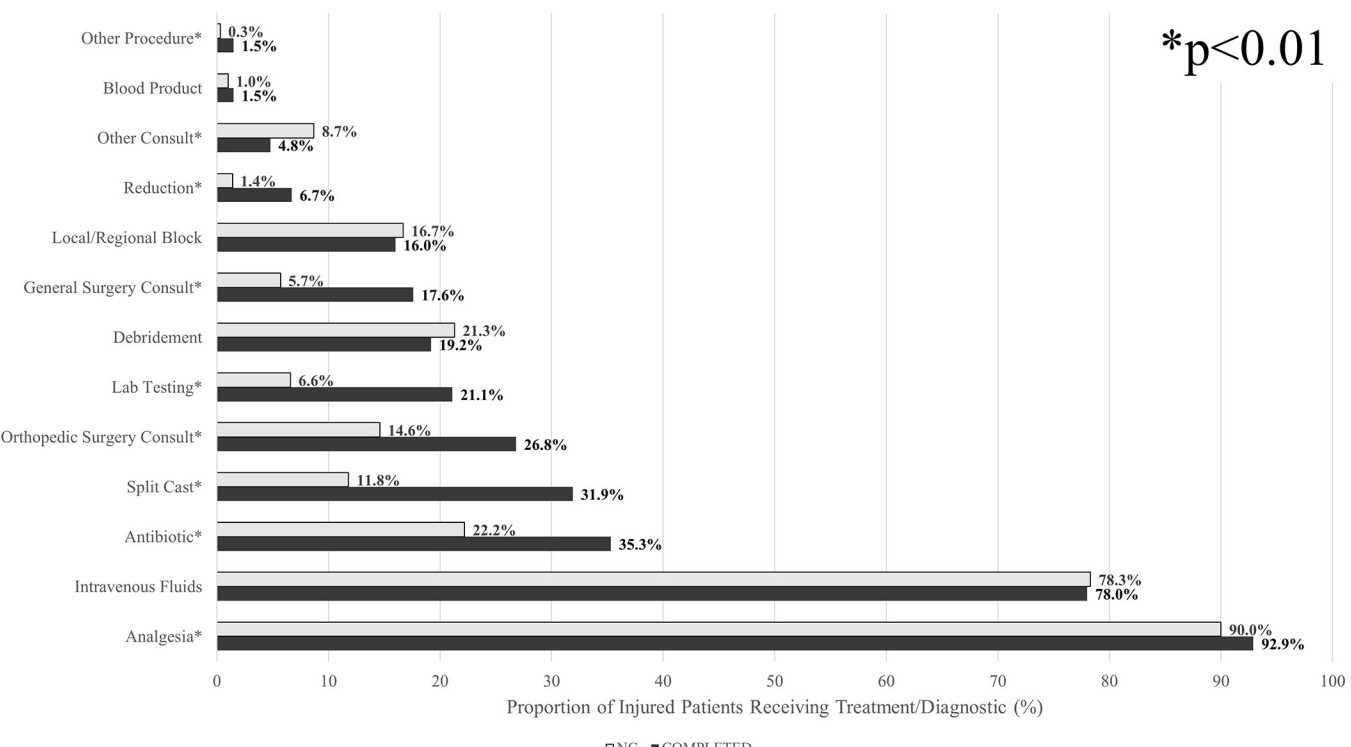

**Fig 3. Care utilization patterns of imaging completion cohorts.**

diagnostic imaging study was 1.1 (IQR 0.7–2.7) hours. Median time to disposition did not differ significantly between cohorts (5.9 (IQR 3.4–13.4) hours COMPLETED vs 6.5 (IQR 3.0–12.7) hours NC, p = 0.11), but time to death was significantly longer in COMPLETED patients (5.7 (IQR 2.1–12.0) hours vs 3.4 (IQR 1.5–6.5) hours, p = 0.01).

## Discussion

In this study, we characterized diagnostic imaging patterns in a prospective, multicenter trauma cohort and found that receiving prescribed diagnostic imaging is associated with increased early survival after injury in Cameroon. To our knowledge, we present the first data demonstrating a positive association between diagnostic imaging and trauma survival in central SSA. However, these data are consistent with the existing literature from developed trauma systems demonstrating improved outcomes among patients who receiving imaging as part of the standard trauma assessment [1,27–29]. Completion of prescribed imaging in our cohort

**Table 3. Associations with emergency department survival.**

|  | Odds Ratio* | 95% CI | p-value |
| --- | --- | --- | --- |
| Highest Estimated Abbreviated Injury Score | 0.11 | 0.09–0.14 | <0.01 |
| Access to Liquid Petroleum Gas | 3.08 | 2.04–4.63 | <0.01 |
| Male Sex | 1.18 | 0.71–1.96 | 0.53 |
| Age | 0.99 | 0.97–1.00 | 0.02 |
| **Received Prescribed Radiologic Study** | **5.00** | **3.32–7.55** | **<0.01** |

*Adjusted for age, sex, socioeconomic status, and injury severity CI, confidence interval.

was also found to be independently associated with increased rates of hospital admission and utilization of additional medical services. Although it is not possible establish a causal link between imaging completion and survival using observational data, we hypothesize that diagnostic imaging facilitates prompt diagnosis and treatment of injuries that might otherwise be missed. These data provide preliminary data to support advocacy efforts to direct available resources toward increasing diagnostic capacity to improve trauma outcomes in Cameroon.

To support development of a context-tailored, data-driven approach to reducing diagnostic delays in Cameroon, we mapped current diagnostic capacity and imaging utilization patterns at each site. Our findings demonstrate that only 37% of injured patients receive diagnostic imaging as part of the trauma assessment. These findings are consistent with the limited existing literature which highlights variable availability and access barriers to radiology in SSA and other LMICs [30,31]. Importantly, radiographs comprised the vast majority of imaging performed but are not sufficiently sensitive to diagnose intrabdominal hemorrhage; only 4.8% of trauma patients received imaging capable of screening for intra-abdominal hemorrhage. Point-of-care ultrasound was not available at any of the study sites during the study period. In Cameroon, our data suggest that both under-prescription and barriers to imaging completion play a role in the limited utilization of radiology. Fewer than half of injured patients are prescribed imaging compared to developed trauma settings where almost all patients receive imaging. Overall, 59.0% of studies performed demonstrate pathology; this high true positive rate strongly indicates that expanding imaging access would likely yield radiographic diagnosis of additional injuries. As a high proportion of patients presenting with trauma mechanisms are discharged home from the emergency department, failing to accurately diagnose injury at the initial trauma evaluation may lead to inappropriate discharge and contribute to out-of-hospital morbidity and mortality. Among those prescribed imaging, cost interference with care was associated with failure to complete radiology studies. Significant demographic differences in SES and rates of past medical contact also suggest cost interference with care. Furthermore, inability to pay was the most frequently reported reason cited for not receiving prescribed studies. Two other common reasons cited, patient preference and departure against medical advice, may also be related to cost barriers to care. Previous work in Cameroon has shown that utilization of formal healthcare services is strongly associated with worse financial consequences after injury [23]. Subsidizing radiology costs may be critical in expanding access to diagnostic imaging. Severe injuries with rapidly fatal progression represented another possible reason for study non-receipt. However, time to death in NC patients (3.4 hours) vs time to diagnostic imaging in COMPLETED patients (1.1 hours) suggests ample time for rapid diagnostic imaging of critically injured patients in an optimized trauma system.

Importantly, reducing radiology costs will need to be balanced with the need to increase access to functional imaging equipment. Lack of functional equipment was the second most reported reason for study non-completion, however, high up-front costs of purchasing and maintaining equipment are current barriers to expansion. The Lancet Commission on Diagnostics recommends use of digitalization with point-of-care diagnostics as a potential solution to improving access to diagnostics [32]. Training primary trauma care providers in eFAST is one possible application of this recommendation with potential to increase diagnostic access and diminish patient cost barriers eFAST is a bedside ultrasound protocol known to decrease time to operative intervention in developed settings. Multiple studies have established a sensitivity between 85–96% and a specificity of 98% for detection of hemoperitoneum and hemopericardium in blunt trauma and is taught in the American College of Surgeon's Advanced Trauma Life Support course [33,34]. Several eFAST features make it a promising modality for adaptation to the LMIC setting. Specifically, eFAST can be performed and interpreted at the patient's bedside by the examining provider, avoiding potential bottlenecks in transport,

administration, and radiologist interpretation. Task shifting of eFAST from radiologists to primary trauma care providers may alleviate access barriers due to limited specialist training [35–38]. Several LMIC studies have demonstrated promising educational efficacy of context-adapted training programs in teaching small clinician cohorts to perform point-of-care ultrasonography, including eFAST, in Rwanda, Nigeria, Sierra Leone and Uganda [39–42]. Conventional ultrasound machines are expensive and scarce in Cameroon, likely contributing to the 2% rate of injured patients who received ultrasound among our cohort. Recent development of low-cost smartphone-based ultrasonography probes could facilitate increased dissemination of eFAST in LMIC but rigorous studies to establish feasibility and effectiveness in the Cameroonian context are needed [43]. eFAST may increase imaging accessibility while avoiding some of the pitfalls of excessive imaging found in high-income countries, which include radiation exposure, increased cost of care, and environmental impact, particularly for low risk and redundant studies [44,45]. Affordable diagnostics with minimal exposure risks and resource expenditure are optimal to maintain an appropriate balance between under- and over-utilization in LMICs. There are currently no uniform practice guidelines routinely followed in Cameroon regarding indications for imaging in trauma patients. Developing contextually appropriate imaging guidelines may provide a target for streamlining evidenced based care and improving outcomes after imaging in Cameroon, while limiting over-prescription of studies.

This study's strengths include its novelty in showing a positive association between diagnostic imaging and trauma survival in central SSA, aligning with existing literature from high-income countries. Additionally, it characterizes reasons for which studies were not completed, delineating possible targets for improving access to diagnostic imaging. Its prospective, multicenter trauma registry cohort without exclusions is another key strength to the data presented.

There are several limitations to this study. Demographic questions rely on patient reporting, which may be unavailable from critically injured patients. In cases of data missingness, percentages were calculated based on the number of patients with data collected for the variable in question. While we present an association between diagnostic imaging and early injury survival, trauma death is a complex outcome that results from the interplay of patient and injury characteristics, physiologic derangement, and management factors. The exact mechanisms underlying these associations cannot be determined using observational data. Markers of higher socioeconomic status, injury severity, and certain injury mechanisms all cluster with imaging completion. Although these variables were adjusted for in our regression model, we acknowledge the potential impact of additional variability that may not be adequately accounted for. Furthermore, LPG is only an indicator and not a comprehensive evaluation of SES. More accurate assessment techniques have been published using DHS and DHS derived methodology but for were beyond the scope of the present manuscript [24]. Finally, there are vast differences between LMIC and it should be noted that generalizability to other healthcare systems will be imperfect.

## Conclusion

Only 37% of injured Cameroonian patients receive diagnostic imaging as part of the trauma assessment. Receiving prescribed diagnostic imaging in the ED is associated with early injury survival in Cameroon. In a resource-limited setting, directing available resources toward increasing diagnostic capacity has potential to facilitate timely diagnosis and treatment of injury and may provide a target to reduce preventable deaths.

## Supporting information

**S1 Data.**
(XLSX)

## Acknowledgments

Alain Chichom-Mefire and S. Ariane Christie are joint senior authors.

## Author Contributions

**Conceptualization:** Matthew Driban, Fanny N. Dissak-Delon, Melissa Carvalho, Mbiarikai Mbianyor, Georges A. Etoundi-Mballa, Thompson Kingue, Richard L. Njock, Daniel N. Nkusu, Jean-Gustave Tsiagadigui, Catherine Juillard, Alain Chichom-Mefire, S. Ariane Christie.

**Data curation:** Matthew Driban, Alain Chichom-Mefire, S. Ariane Christie.

**Formal analysis:** Matthew Driban, S. Ariane Christie.

**Funding acquisition:** Matthew Driban, Catherine Juillard, Alain Chichom-Mefire, S. Ariane Christie.

**Investigation:** Matthew Driban, Alain Chichom-Mefire, S. Ariane Christie.

**Methodology:** Catherine Juillard, Alain Chichom-Mefire, S. Ariane Christie.

**Supervision:** S. Ariane Christie.

**Writing – original draft:** Matthew Driban.

**Writing – review & editing:** Matthew Driban, Fanny N. Dissak-Delon, Melissa Carvalho, Mbiarikai Mbianyor, Juan C. Puyana, Catherine Juillard, Alain Chichom-Mefire, S. Ariane Christie.

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
