## [Decision Letter · Decision Letter 0]

14 Mar 2023

PGPH-D-22-02121

Failure to Receive Prescribed Imaging Portends Increased Early Mortality after Injury in Cameroon

Dear Dr. Driban,

Thank you for submitting your manuscript to PLOS Global Public Health. After careful consideration, we feel that it has merit but does not fully meet PLOS Global Public Health’s publication criteria as it currently stands. Therefore, we invite you to submit a revised version of the manuscript that addresses the points raised during the review process.

Please address the points raised by the reviewers. Specifically, focus on the issues of the inclusion criteria and the analyses reported as they were shared points of concern for both reviews.

We look forward to receiving your revised manuscript.

Kind regards,

Hani Mowafi, M.D., M.P.H.

Academic Editor

Journal Requirements:

1. In the ethics statement in the Methods, you have specified that verbal consent was obtained. Please provide additional details regarding how this consent was documented and witnessed, and state whether this was approved by the IRB

2. Please send a completed 'Competing Interests' statement, including any COIs declared by your co-authors. If you have no competing interests to declare, please state "The authors have declared that no competing interests exist". Otherwise please declare all competing interests beginning with the statement "I have read the journal's policy and the authors of this manuscript have the following competing interests:"

3. Please amend your detailed Financial Disclosure statement. This is published with the article. It must therefore be completed in full sentences and contain the exact wording you wish to be published.

4. We do not publish any copyright or trademark symbols that usually accompany proprietary names, eg  ©, ®, ™  (e.g. next to drug or reagent names). Please remove all instances of trademark/copyright symbols throughout the text, including ® on page 15.

5. In the online submission form, you indicated that "Data available upon request from the authors". All PLOS journals now require all data underlying the findings described in their manuscript to be freely available to other researchers, either 1. In a public repository, 2. Within the manuscript itself, or 3. Uploaded as supplementary information.

Additional Editor Comments (if provided):

Reviewers' comments:

Reviewer's Responses to Questions

**Comments to the Author**

1. Does this manuscript meet PLOS Global Public Health’s publication criteria? Is the manuscript technically sound, and do the data support the conclusions? The manuscript must describe methodologically and ethically rigorous research with conclusions that are appropriately drawn based on the data presented.

Reviewer #1: Yes

Reviewer #2: Yes

2. Has the statistical analysis been performed appropriately and rigorously?

Reviewer #1: Yes

Reviewer #2: Yes

3. Have the authors made all data underlying the findings in their manuscript fully available (please refer to the Data Availability Statement at the start of the manuscript PDF file)?

Reviewer #1: No

Reviewer #2: Yes

4. Is the manuscript presented in an intelligible fashion and written in standard English?

Reviewer #1: Yes

Reviewer #2: Yes

5. Review Comments to the Author

Reviewer #1: I wish to thank the authors for an excellent incursion into issues we face with diagnostics in Variable resource settings everyday. This paper puts into scientific format much of what the LMIC trauma care practitioner can see every day- that there is need to address issues of access to diagnostics to improve morbidity and mortality from trauma and other non-communicable and communicable diseases and injuries.

A few issues to be addressed-

1. "70 reportedly at least one government-funded imaging center in each of Cameroon’s 10 geopolitical 71 regions offers computed tomography, imaging capacity is largely concentrated in three major 72 urban areas: Yaoundé, Douala, and Bafoussam. Over 80% of Cameroonian imaging centers are 73 limited to radiographs and/or ultrasound. Most existing equipment is donated, refurbished, or has 74 been in use for over 10 years." The above statement requires a citation (Lines 70-74).

2. Ethics Statement "Verbal consent was obtained by research assistant" (Line 97) This raises some concern of data safety. What is the procedure for obtaining consent for use of the registry? It is unlikely that it would be standard procedure to obtain verbal permission from Cameroon NEC, and UCLA Institutional Review Boards for use of this data. Authors should kindly review this, and provide an exemption IRB certificate or documented Ethics Approval at the least. If no formal permission has been obtained, the authors should consider a retrospective approval so as to confirm that local partners are agreed to the use of this data. This is a major concern affecting the reviewer decision.

3. Lines 111-112- "access to liquid petroleum gas (LPG) was used as a surrogate indicator of higher socioeconomic status" The limitations of using this should be highlighted in limitations as more accurate measures of SES could have been collected.

4. Line 135 Table 1. Demographics of Injured Patients Authors suggest missing demographics data in limitations but no recourse is made to how this was handled. Are these percentages of the total (n)- in which case, no demographics data is missing- or percentages of varying (n)s based on data is available for each demographic consideration? Authors to please clarify in the text or in the case of handling of missingness, in the table.

5. Line 136 IQR, interquartile range. IQR is not referenced in table 1 and should not be in the footnotes

6. 173 to 174- The authors noted that "NC patients reported significantly higher rate of cost interference with care (60.5%) compared to COMPLETED" Could this be because patients could afford the investigations? A higher capacity to pay results in better care and not just in receipt of more imaging which implies a likelihood to progress in care. As NC patients reported a "significantly higher rate of cost interference with care (60.5%) compared to COMPLETED patients" Line 181- "NC patients left against medical advice (35.0% vs. 21.3%, p<0.01) or died (5.6% vs. 1.3%, 182 p<0.01) more frequently" Authors should look into the sequence- did NC have less imaging because they died before imaging was possible? or left against medical advice before imaging was available? Or physicians considered it medical futility to carry out some investigations in these patients? Is there a threshold of days on admission considered, as a subset of the analysis to ensure that early deaths do not confound these results? Can this be emphasized? Kindly add in a time to death (in NC) versus time to intervention in C to clarify.

7. Line 185 "Time to disposition did not differ significantly between cohorts." In consideration of significance- Even if it did not differ significantly, did time to disposition differ at all? Kindly state the difference if any (means)and p-value. Perhaps soem issues are not statistically significant but might be clinically significant.

8. Authors should confirm- Is there the possibility that point of care focused assessment sonography for trauma might have been missed in the registry records? Bedside/ED ultrasound scan in some LMICs, by experience, is usually not ordered or documented as a radiological "procedure" or standard assessment.

10. Lines 211-212 "In Cameroon, our data suggest that both under-prescription and barriers to imaging completion play a role in the limited utilization of radiology." Underprescription does not appear to be the focus of this study, and should not be emphasized.

11. The Limitations are excellent observations. Generalizability to other healthcare systems *will* be imperfect, and should be acknowledged as such

12. Authors should consider including a reference tot he Lancet Commission on Diagnostics as there is much alignment with the intent of this paper- and for suggestions of set standards

13. Please edit reference 10 (Line 296)

*14. Figure 1- showing what was not completed in Figure 1 would help clarify what specifically is NC (to mirror the completed. Kindly add this in for more clarity on the part of the reader.*

15. Authors should also kindly briefly comment of overutilization of imaging with financial and environmental implications that typically can occur in high resource settings, and where the balance should be, in the discussion.

16. Finally, the authorship order raises a few concerns of LMIC authors being "stuck in the middle." Could authors consider co-first authorship based on other contribution considerations, co-senior authorship, or at the minimum, having a Cameroonian corresponding author, as this work is done on data from Cameroon? there is some reflexivity required here, and the authorship team would do well to consider carefully the inclusive placement of LMIC authors in the hierarchy of authorship.

Kindly consider the data availability policy of Plos Global Public Health https://journals.plos.org/globalpublichealth/s/submission-guidelines#loc-data-reporting

*Thank you again for this insightful work and I look forward to the addressing of these observations.*

*Reviewer #2: This study evaluated patterns of imaging performance in a prospective multi-site trauma registry cohort in Cameroon in 2017-2019. The study has valuable information and I read it with interest, but some comments should be addressed prior to moving to the next step. My comments are as follows:*

Major comments:

• While in the text of the manuscript, it was mentioned correctly that findings are associations, the title of the manuscript mistakenly shows a causality: “Failure to Receive Prescribed Imaging Portends Increased Early Mortality after Injury in Cameroon”, “increased” should be replaced with “associated with increased”

• Methods: “Verbal consent was obtained by research assistants.” It is not clear how verbal consent was obtained as participants were injured and usually it is not possible to take consent from people who suffer from a recent injury, also if for a participant the outcome of the injury was death, how was the consent obtained?

• Methods: In this study, data was collected on paper registry forms which were then entered into a cloud-based server. The information about data collection was limited but as I understood this form was a usual history-taking form designed for trauma patients and the data was collected from patients as a routine for the personnel, while they were admitted and medical procedures were on-process. So, the quality of the data is questionable. Especially, it may be biased for underreporting from severely injured patients. For example, about the results of table 1: maybe severely injured patients did not complete the imaging because maybe their hemodynamic status was unstable and they were unable to perform imaging and so due to their instability maybe they did not answer having access to LPG or having a cell phone or having a past medical history. Authors should clarify how they addressed these possibilities of bias.

• Methods: For adjusting logistic regression, it should be stated how the injury severity was assessed. Also, who did assess the severity?

• Please add strengths of the study prior to its limitations.

• Insurance has a great impact on healthcare utilization. In this study, insurance coverage status was not investigated and also was not mentioned in the discussions.

Minor comments:

• The manuscript did not mention the references for physicians defining the indications of radiologic imaging requests that were authorized or published by Cameroon’s ministry of health or other related health sectors. If there are no local guidelines, the usual references for physicians should be mentioned briefly.

• In the background section was mentioned: “As part of a larger quality improvement initiative and to assist priority setting for policymakers, we evaluated trauma outcomes among patients who did and did not receive indicated imaging in the Emergency Department (ED). We hypothesize that receiving imaging is associated with increased early injury survival.” While these two sentences provided valuable information, I recommend presenting exactly and clearly the aim of the study, instead.

*• Introduction: “Over 80% of Cameroonian imaging centers are limited to radiographs and/or ultrasound.” This sentence needs a reference.*

*6. PLOS authors have the option to publish the peer review history of their article (what does this mean?). If published, this will include your full peer review and any attached files.*

**Do you want your identity to be public for this peer review?** *For information about this choice, including consent withdrawal, please see our Privacy Policy.*

*Reviewer #1: No*

*Reviewer #2: No*

**

*While revising your submission, please upload your figure files to the Preflight Analysis and Conversion Engine (PACE) digital diagnostic tool, https://pacev2.apexcovantage.com/. PACE helps ensure that figures meet PLOS requirements. To use PACE, you must first register as a user. Registration is free. Then, login and navigate to the UPLOAD tab, where you will find detailed instructions on how to use the tool. If you encounter any issues or have any questions when using PACE, please email PLOS at figures@plos.org. Please note that Supporting Information files do not need this step.*

---

## [Decision Letter · Decision Letter 1]

4 Jul 2023

PGPH-D-22-02121R1

Failure to Receive Prescribed Imaging is Associated with Increased Early Mortality after Injury in Cameroon

Dear Dr. Driban,

Thank you for submitting your manuscript to PLOS Global Public Health. After careful consideration, we feel that it has merit but does not fully meet PLOS Global Public Health’s publication criteria as it currently stands. Therefore, we invite you to submit a revised version of the manuscript that addresses the points raised during the review process.

Your team's work is very thorough, and I commend the authors on the depth and breadth of this revision. The rebuttal was extremely clear and well done. All (initially extensive) concerns warranting a major revision were effectively addressed as noted by both reviewers.

1. In response to your desire that Alain Chichom-Mefire and S. Ariane Christie be joint senior authors, kindly add a note in your acknowledgements to this effect- "Alain Chichom-Mefire and S. Ariane Christie are joint senior authors." This is the only change needed to move this forward.

2. In view of the singular concern by Reviewer #2, I can confirm that the location of the supplementary file named "CTR_Radiology_PLOS_Deidentified.xls" mentioned in your response is at the end of the revised document following the figures and just before the manuscript with tracked changes. Thank you for including this. As the reviewer noted, this is particularly important to ensure the transparency and reproducibility of your work. 

Again, thank you for your extensive work on improving the manuscript through such a thoughtful revision.

We look forward to receiving your revised manuscript.

Kind regards,

Barnabas Tobi Alayande

Academic Editor

Journal Requirements:

2. We have noticed that you have uploaded Supporting Information files, but you have not included a list of legends. Please add a full list of legends for your Supporting Information files after the references list.

Additional Editor Comments (if provided):

Reviewers' comments:

Reviewer's Responses to Questions

**Comments to the Author**

1. If the authors have adequately addressed your comments raised in a previous round of review and you feel that this manuscript is now acceptable for publication, you may indicate that here to bypass the “Comments to the Author” section, enter your conflict of interest statement in the “Confidential to Editor” section, and submit your "Accept" recommendation.

Reviewer #1: All comments have been addressed

Reviewer #2: All comments have been addressed

2. Does this manuscript meet PLOS Global Public Health’s publication criteria? Is the manuscript technically sound, and do the data support the conclusions? The manuscript must describe methodologically and ethically rigorous research with conclusions that are appropriately drawn based on the data presented.

Reviewer #1: Yes

Reviewer #2: Yes

3. Has the statistical analysis been performed appropriately and rigorously?

Reviewer #1: Yes

Reviewer #2: Yes

4. Have the authors made all data underlying the findings in their manuscript fully available (please refer to the Data Availability Statement at the start of the manuscript PDF file)?

Reviewer #1: Yes

Reviewer #2: Yes

5. Is the manuscript presented in an intelligible fashion and written in standard English?

Reviewer #1: Yes

Reviewer #2: Yes

6. Review Comments to the Author

Reviewer #1: All comments have been addressed, and I would like to thank the authors for a very thorough job well done. I am particularly impressed by your willingness to consider a more equitable author distribution and have co-senior authors from both contexts acknowledged in their authorship list.

Reviewer #2: I have reviewed the revised version of your manuscript titled "Failure to Receive Prescribed Imaging is Associated with Increased Early Mortality after Injury in Cameroon" and I am pleased to see that you have addressed all of my previous comments. I appreciate the effort you have put into revising the manuscript.

However, I would like to bring to your attention that I could not locate the supplementary file entitled "CTR_Radiology_PLOS_Deidentified.xls" as mentioned in your response. It is essential that all supporting data and files are provided to ensure the transparency and reproducibility of your research. I kindly request that you double-check the supplementary files and ensure that they are properly included with the revised submission.

Overall, I commend your thoroughness in addressing the previous concerns raised by the reviewers, and I believe the manuscript has significantly improved as a result. Once the issue with the supplementary file is resolved, I think that the manuscript will be ready for publication.

7. PLOS authors have the option to publish the peer review history of their article (what does this mean?). If published, this will include your full peer review and any attached files.

**Do you want your identity to be public for this peer review?** For information about this choice, including consent withdrawal, please see our Privacy Policy.

Reviewer #1: No

Reviewer #2: No

---

## [Editor Report · Decision Letter 2]

25 Jul 2023

Failure to Receive Prescribed Imaging is Associated with Increased Early Mortality after Injury in Cameroon

PGPH-D-22-02121R2

Dear Mr. Driban,

We are pleased to inform you that your manuscript 'Failure to Receive Prescribed Imaging is Associated with Increased Early Mortality after Injury in Cameroon' has been provisionally accepted for publication in PLOS Global Public Health.

Best regards,

Barnabas Tobi Alayande

Academic Editor